# Temporal Logic Imitation: Learning Plan-Satisficing Motion Policies from Demonstrations

**Yanwei Wang**
MIT

**Nadia Figueroa**
University of Pennsylvania

**Shen Li**
MIT

**Ankit Shah**
Brown University

**Julie Shah**
MIT

**Abstract:** Learning from demonstration (LfD) has successfully solved tasks featuring a long time horizon. However, when the problem complexity also includes human-in-the-loop perturbations, state-of-the-art approaches do not guarantee the successful reproduction of a task. In this work, we identify the roots of this challenge as the failure of a learned continuous policy to satisfy the discrete plan implicit in the demonstration. By utilizing modes (rather than subgoals) as the discrete abstraction and motion policies with both mode invariance and goal reachability properties, we prove our learned continuous policy can simulate any discrete plan specified by a linear temporal logic (LTL) formula. Consequently, an imitator is robust to both task- and motion-level perturbations and guaranteed to achieve task success. **Project page:** https://sites.google.com/view/ltl-ds

**Keywords:** Certifiable Imitation Learning, Dynamical Systems, Formal Methods

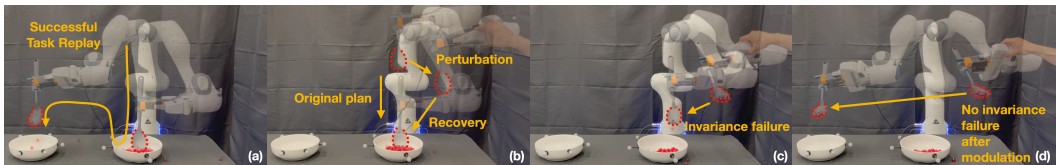

Figure 1: **(a)** A successful replay of the scooping task. The robot is **(b)** robust to motion-level perturbations; **(c)** experiences an invariance failure (i.e., drops material) after a task-level perturbation; and **(d)** re-scoops after a task-level perturbation, avoiding failure after DS motion policy modulation.

## 1 Introduction

In prior work, learning from demonstration (LfD) [1, 2] has successfully enabled robots to accomplish multi-step tasks by segmenting demonstrations (primarily of robot end-effector or tool trajectories) into sub-tasks/goals [3, 4, 5, 6, 7, 8], phases [9, 10], keyframes [11, 12], or skills/primitives/options [13, 14, 15, 16]. Most of these abstractions assume reaching subgoals sequentially will deliver the desired outcomes; however, successful imitation of many manipulation tasks with spatial/temporal constraints cannot be reduced to imitation at the motion level unless the learned motion policy also satisfies these constraints. This becomes highly relevant if we want robots to not only imitate, but also generalize, adapt and be robust to perturbations imposed by humans who are in the loop of task learning and execution. LfD techniques that learn stable motion policies with convergence guarantees (e.g., Dynamic Movement Primitives (DMP) [17], Dynamical System (DS) [18]) are capable of providing such desired properties but only at the motion level. As shown in Fig. 1 (a-b) the robot can successfully replay a soup-scooping task while being robust to physical perturbations with a learned DS. Nevertheless, if the spoon orientation is perturbed to a state where the material is dropped, Fig. 1 (c), the motion policy will still lead the robot to the target, unaware of the task-level failure or how to recover from it. To alleviate this, in this work, we introduce an imitation learning approach that is capable of i) reacting to such task-level failures with Linear Temporal Logic (LTL) specifications, and ii) modulating the learned DS motion policies to avoid repeating those failures as shown in Fig. 1 (d).

**Example** We demonstrate that successfully reaching a goal via pure motion-level imitation does not imply successful task execution. The illustrations in Fig. 2 represent a 2D simplification of the soup-scooping task, where task success requires a continuous trajectory to simulate a discrete plan of consecutive transitions through the colored regions. Human demonstrations, shown in Fig. 2 **(a)**, are employed to learn a DS policy [19], depicted by the streamlines in Fig. 2 **(b)**. The policy is

6th Conference on Robot Learning (CoRL 2022), Auckland, New Zealand.

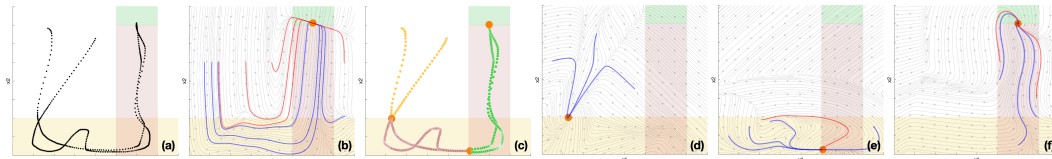

Figure 2: A mode abstraction of the 2D soup-scooping task: $x_1$ is the spoon's orientation, and $x_2$ is spoon's distance to the soup. **(a)** Task: To move the spoon's configuration from white region (spoon without soup) $\Rightarrow$ yellow region (spoon in contact with soup) $\Rightarrow$ pink region (spoon holding soup) $\Rightarrow$ green region (soup at target). (Note that transitions (white $\Rightarrow$ pink) and (white $\Rightarrow$ green) are not physically realizable.) Black curves denote successful demonstrations. **(b)** Learning DS policies [19] over unsegmented data can result in successful task replay (blue trajectories), but lacks a guarantee due to invalid transitions (red trajectories). **(c)** Trajectories segmented into three colored regions (modes) with orange attractors. **(d-f)** Learning DSs on segments may still result in *invariance failures* (i.e., traveling outside of modes as depicted by red trajectories).

stress-tested by applying external perturbations, displacing the starting states of the policy rollouts. As shown, only blue trajectories succeed in the task, while the red ones fail due to discrete transitions that are not physically realizable (e.g., white $\Rightarrow$ pink). As shown in Fig. 2 **(c-f)**, even if the demonstrations are further segmented by subgoals (and corresponding DS policies are learned), this issue is not mitigated. While one could treat this problem as covariate shift and solve it by asking the humans for more demonstrations [20], in this work, we frame it as the mismatch between a learned continuous policy and a discrete task plan specified by the human in terms of a logical formula. Specifically, the core challenges illustrated by this example are two-fold: 1) subgoals only impose point constraints that are insufficient to represent the boundary of a discrete abstraction; and 2) the continuous policy can deviate from a demonstrated discrete plan when perturbed to unseen parts of the state space, and is incapable of replanning to ensure all discrete transitions are valid.

To address these challenges, our proposed approach employs "modes" as the discrete abstractions. We define a *mode* as a set of robot and environment configurations that share the same sensor reading [21, 22]; e.g., in Fig. 2 each colored region is a unique mode, and every mode has a boundary that imposes path constraints on motion policies. Additionally, we use a task automaton as a receding-horizon controller that replans when a perturbation causes the system to travel outside a mode boundary and triggers an unexpected sensor change; e.g., detecting a transition from yellow $\Rightarrow$ white instead of the desired yellow $\Rightarrow$ pink will result in a new plan: white $\Rightarrow$ yellow $\Rightarrow$ pink $\Rightarrow$ green. In this work, we synthesize a task automaton from a linear temporal logic formula (LTL) that specifies all valid mode transitions. We denote the problem of learning a policy that respects these mode transitions from demonstrations as *temporal logic imitation* (TLI). In contrast to temporal logic planning (TLP) [23], where the workspace is typically partitioned into connected convex cells with known boundaries, we do not know the precise boundaries of modes; consequently, the learned policy might prematurely exit the same mode repeatedly, causing the task automaton to loop without termination. To ensure any discrete plan generated by the automaton is feasible for the continuous policy, the bisimulation criteria [24, 25] must hold for the policy associated for each mode. Specifically, every state starting in the same mode should stay in the mode **(invariance)** until eventually reaching the next mode **(reachability)**. The violations of these conditions are referred to as *invariance failures* and *reachability failures* respectively.

**Contributions** First, we investigate TLP in the setting of LfD, and introduce TLI as a novel formulation to address covariate shift by proposing imitation with respect to a mode sequence instead of a motion sequence. Second, leveraging modes as the discrete abstraction, we prove that a state-based continuous behavior cloning (BC) policy with a global stability guarantee can be modulated to simulate any LTL-satisfying discrete plan. Third, we demonstrate that our approach LTL-DS, adapts to task-level perturbations via a LTL-satisfying automaton's replanning and recovers from motion-level perturbations via DS' stability during a multi-step, non-prehensile manipulation task.

## 2 Related Works

**Temporal Logic Motion Planning** LTL is a task specification language widely used in robot motion planning [26, 27, 28, 23]. Its ease of use and efficient conversion [29] to an automaton have spurred substantial research into TLP [25, 30, 31], which studies how to plan a continuous trajectory that satisfies a LTL formula. However, TLP typically assumes known workspace partitioning and boundaries *a priori*, both of which are unknown in the rarely explored TLI setting. While a robot can still plan in uncertain environments [32, 33], LfD bypasses the expensive search in high-dimensional space. Recent works [34, 35] have considered temporal logic formulas as side-information to demonstrations, but these formulas are treated as additional loss terms or rewards,

and are not guaranteed to be satisfied. The key motivation for using LTL is to generate a reactive discrete plan, which can also be achieved by a finite state machine [14] or behavior tree [36].

**Behavior Cloning** We consider a subclass of LfD methods called state-based behavior cloning (BC) that learns the state-action distribution observed during demonstrations [37]. DAGGER [20], a BC-variant fixing covariate shift, could reduce the invariance failures depicted in Fig. 2, but requires online data collection, which our framework avoids with a LTL specification. To satisfy goal reachability, we employ a DS-based LfD technique [38]. Alternatives to this choice include certified NN-based methods [39, 40], DMPs [41], partially contracting DS [42] and Euclideanizing-flows [43]. To satisfy mode invariance, we modulate the learned DS to avoid invariance failure as state-space boundaries [44], similar to how barrier functions are learned to bound a controller [45, 46, 47].

**Multi-Step Manipulation** Prior LfD works [13, 14, 10, 48] tackle multi-step manipulation by segmenting demonstrations via a hidden Markov model. Using segmented motion trajectories, [13] learned a skill tree, [14] learned DMPs, [10] learned phase transitions, and [49] learned a task model. Most of these works assume a linear sequence of prehensile subtasks (pick-and-place) without considering how to replan when unexpected mode transitions happen. [48, 49] considered a non-prehensile scooping task similar to ours, but their reactivity only concerned collision avoidance in a single mode. [50, 6] improved BC policies with RL, but offered no guarantee of task success.

## 3   Temporal Logic Imitation: Problem Formulation

Let $x \in \mathbb{R}^n$ represent the $n$-dimensional continuous state of a robotic system; e.g., the robot's end-effector state in this work. Let $\alpha = [\alpha_1, ..., \alpha_m]^T \in \{0, 1\}^m$ be an $m$-dimensional discrete sensor state that uniquely identifies a mode $\sigma = \mathcal{L}(\alpha)$. We define a system state as a tuple, $s = (x, \alpha) \in \mathbb{R}^n \times \{0, 1\}^m$. Overloading the notation, we use $\sigma \in \Sigma$, where $\Sigma = \{\sigma_i\}_{i=1}^{\mathcal{M}}$, to represent the set of all system states within the same mode—i.e., $\sigma_i = \{s = (x, \alpha) \mid \mathcal{L}(\alpha) = \sigma_i\}$. In contrast, we use $\delta_i = \{x | s = (x, \alpha) \in \sigma_i\}$ to represent the corresponding set of robot states. Note $x$ cannot be one-to-one mapped to $s$, e.g., a level spoon can be either empty or holding soup. Each mode is associated by a goal-oriented policy, with goal $x_i^* \in \mathbb{R}^n$. A successful policy that accomplishes a multi-step task $\tau$ with a corresponding LTL specification $\phi$ can be written in the form:

$$\dot{x} = \pi(x, \alpha; \phi) = \Sigma_{i=1}^{\mathcal{M}} \delta_{\Omega_\phi(\alpha)\sigma_i} f_i(x; \theta_i, x_i^*) \tag{1}$$

with $\delta_{\Omega_\phi(\alpha)\sigma_i}$ being the Kronecker delta that activates a mode policy $f_i(x; \theta_i, x_i^*) : \mathbb{R}^{n+m} \to \mathbb{R}^n$ encoded by a set of learnable parameters $\theta_i$ and goal $x_i^*$. Mode activation is guided by an LTL-equivalent automaton $\Omega_\phi(\alpha) \to \sigma_i$ choosing mode $\sigma_i$ based on current sensor reading $\alpha$.

**Demonstrations** Let demonstrations for a task $\tau$ be $\Xi = \{\{x^{t,d}, \dot{x}^{t,d}, \alpha^{t,d}\}_{t=1}^{T_d}\}_{d=1}^{D}$ where $x^{t,d}, \dot{x}^{t,d}, \alpha^{t,d}$ are robot state, velocity, and sensor state at time $t$ in demonstration $d$, respectively, and $T_d$ is the length of each $d$-th trajectory. A demonstration is successful if the continuous motion traces through a sequence of discrete modes that satisfies the corresponding LTL task specification.

**Perturbations** External perturbations, which many works in Sec. 2 avoid, constitute an integral part of our task complexity. Specifically, we consider: (1) motion-level perturbations that displace the continuous motion within the same mode, and (2) task-level perturbations that drive the robot outside of the current mode. Critically, motion-level perturbations do not cause a plan change instantaneously, but they can lead to future unwanted mode transitions due to covariate shift. Environmental stochasticity is ignored, as its cumulative effects can also be simulated by external perturbations.

**Problem Statement** Given (1) an LTL formula $\phi$ specifying valid mode transitions for a task $\tau$, and (2) successful demonstrations $\Xi$, we seek to learn a policy defined in Eq. 1 that generates continuous trajectories guaranteed to satisfy the LTL specification despite arbitrary external perturbations.

## 4   Preliminaries

### 4.1   LTL Task Specification

LTL formulas consist of atomic propositions (AP), logical operators, and temporal operators [51, 23]. Let $\Pi$ be a set of Boolean variables; an infinite sequence of truth assignments to all APs in $\Pi$ is called the trace $[\Pi]$. The notation $[\Pi], t \models \phi$ means the truth assignment at time $t$ satisfies the LTL formula $\phi$. Given $\Pi$, the minimal syntax of LTL can be described as:

$$\phi ::= p \mid \neg\phi_1 \mid \phi_1 \vee \phi_2 \mid \mathbf{X}\phi_1 \mid \phi_1\mathbf{U}\phi_2 \tag{2}$$

where $p$ is any AP in $\Pi$, and $\phi_1$ and $\phi_2$ are valid LTL formulas constructed from $p$ using Eq. 2. The operator $\mathbf{X}$ is read as 'next,' and $\mathbf{X}\phi_1$ intuitively means the truth assignment to APs at the next time step sets $\phi_1$ as true. $\mathbf{U}$ is read as 'until' and, intuitively, $\phi_1\mathbf{U}\phi_2$ means the truth assignment to APs

sets $\phi_1$ as true until $\phi_2$ becomes true. Additionally, first-order logic operators $\neg$ (not), $\wedge$ (and), $\vee$ (or), and $\rightarrow$ (implies), as well as higher-order temporal operators $\mathbf{F}$ (eventually), and $\mathbf{G}$ (globally), are incorporated. Intuitively, $\mathbf{F}\phi_1$ means the truth assignment to APs eventually renders $\phi_1$ true and $\mathbf{G}\phi_1$ means truth assignment to APs renders $\phi_1$ always true from this time step onward.

## 4.2 Task-Level Reactivity in LTL

To capture the reactive nature of a system given sensor measurements, the *generalized reactivity (1)* (GR(1)) fragment of LTL [29, 30] can be used. Let the set of all APs be $\Pi = \mathcal{X} \cup \mathcal{Y}$, where sensor states form environment APs $\mathcal{X} = \{\alpha_1, ..., \alpha_m\}$ and mode symbols form system APs $\mathcal{Y} = \{\sigma_1, ..., \sigma_l\}$. A GR(1) formula is of the form $\phi = (\phi_e \rightarrow \phi_s)$ [29], where $\phi_e$ models the assumed environment behavior and $\phi_s$ models the desired system behavior. Specifically,

$$\phi_e = \phi_i^e \wedge \phi_t^e \wedge \phi_g^e, \qquad \phi_s = \phi_i^s \wedge \phi_t^s \wedge \phi_g^s \qquad (3)$$

$\phi_i^e$ and $\phi_i^s$ are non-temporal Boolean formulas that constrain the initial truth assignments of $\mathcal{X}$ and $\mathcal{Y}$ (e.g., the starting mode). $\phi_t^s$ and $\phi_t^e$ are LTL formulas categorized as safety specifications that describe how the system and environment should always behave (e.g., valid mode transitions). $\phi_g^s$ and $\phi_g^e$ are LTL formulas categorized as liveness specifications that describe what goal the system and environment should eventually achieve (e.g., task completion) [23]. The formula $\phi$ guarantees the desired system behavior specified by $\phi_s$ if the environment is *admissible*—i.e., $\phi_e$ is true—and can be converted to an automaton $\Omega_\phi$ that plans a mode sequence satisfying $\phi$ by construction [30].

## 4.3 Motion-Level Reactivity in DS

LPV-DS [19] can be learned in minutes from as few as a single demonstration and has form:

$$\dot{x} = f(x) = \sum_{k=1}^{K} \gamma_k(x)(A^k x + b^k) \quad (4) \qquad \begin{cases} (A^k)^T P + P A^k = Q^k, Q^k = (Q^k)^T \prec 0 \\ b^k = -A^k x^* \end{cases} \quad \forall k \quad (5)$$

where $A^k \in \mathbb{R}^{n \times n}$, $b^k \in \mathbb{R}^n$ are the k-th linear system parameters, and $\gamma_k(x) : \mathbb{R}^n \rightarrow \mathbb{R}^+$ is the mixing function. To certify global asymptotic stability (G.A.S.) of Eq. 4, a Lyapunov function $V(x) = (x - x^*)^T P(x - x^*)$ with $P = P^T \succ 0$, is used to derive the stability constraints in Eq. 5. Minimizing the fitting error of Eq. 4 with respect to demonstrations $\Xi$ subject to constraints in Eq. 5 yields a non-linear DS with a stability guarantee [19]. To learn the optimal number $K$ and mixing function $\gamma_k(x)$ we use the Bayesian non-parametric GMM fitting approach presented in [19].

## 4.4 Bisimulation between Discrete Plan and Continuous Policy

To certify a continuous policy will satisfy a LTL formula $\phi$, one can show the policy can simulate any LTL-satisficing discrete plan of mode sequence generated by $\Omega_\phi$. To that end, every mode's associated policy must satisfy the following bisimulation conditions [25, 23]:

**Condition 1 (Invariance).** *All states starting in a mode must remain within the same mode when following that mode's policy; i.e., $\forall i \, \forall t \, (s^0 \in \sigma_i \rightarrow s^t \in \sigma_i)$*

**Condition 2 (Reachability).** *All states starting in a mode must reach their next modes while following the current mode's policy; i.e., $\forall i \, \exists T \, (s^0 \in \sigma_i \rightarrow s^T \in \sigma_j)$*

# 5 LTL-DS: Methodology

To solve the TLI problem in Sec. 3, we introduce a mode-based imitation policy—LTL-DS:

$$\dot{x} = \pi(x, \alpha; \phi) = \underbrace{\Sigma_{i=1}^{\mathcal{M}} \delta_{\Omega_\phi(\alpha)\sigma_i}}_{\text{offline learning}} \underbrace{M_i\big(x; \Gamma_i(x), x_i^*\big)}_{\text{online learning}} \underbrace{f_i(x; \theta_i, x_i^*)}_{\text{offline learning}}, \qquad (6)$$

During offline learning, we synthesize the automaton $\Omega_\phi$ from $\phi$ as outlined in Sec. 4.2 and learn DS policies $f_i$ from $\Xi$ according to Sec. 4.3. While the choice of DS satisfies the reachability condition as explained later, DS rollouts are not necessarily bounded within any region. Neither do we know mode boundaries in TLI. Therefore, an online learning phase is necessary where for each mode policy $f_i$ we learn an implicit function, $\Gamma_i(x) : \mathbb{R}^n \rightarrow \mathbb{R}^+$, that inner-approximates the mode boundary in the state-space of the robot $x \in \mathbb{R}^N$. With a learned $\Gamma_i(x)$ for each mode, we can construct a modulation matrix $M_i$ that ensures each $M_i f_i$ to be mode invariant as discussed below.

## 5.1 Offline Learning Phase

In the offline learning phase, the user provides successful demonstrations $\Xi$ and a LTL formula $\phi$ as described in Sec. 4.2, which necessarily includes the definition of sensors and AP regions.

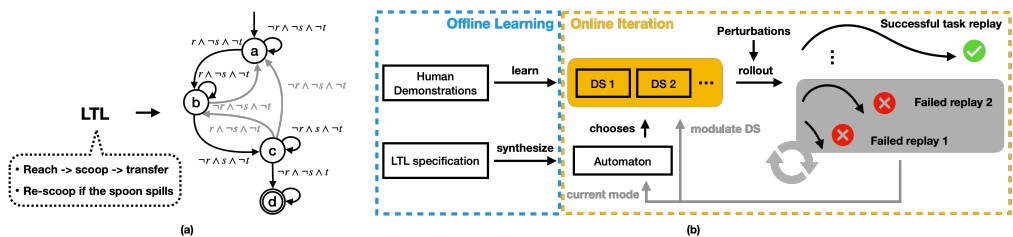

Figure 3: **(a)** Task automaton for a scooping task LTL. Mode $a, b, c, d$ are reaching, scooping, transporting and done mode respectively. Atomic proposition $r, s, t$ denote sensing the spoon reaching the soup, soup on spoon and task success respectively. During successful demonstration, only black mode transition $a \Rightarrow b \Rightarrow c \Rightarrow d$ is observed. Additional gray valid transitions $b \Rightarrow a$, $c \Rightarrow a$ and $c \Rightarrow b$ are given by the LTL to help recover from unexpected mode transitions. **(b)** System flowchart of LTL-DS.

**Synthesis of LTL-Satisficing Automaton** Given $\phi$, we use [52] to convert it into an automaton, which plans a mode sequence that satisfies $\phi$ by construction. Assuming all possible initial conditions for the system are specified in the LTL, the automaton is always deployed from a legal state.
**Sensor-based Motion Segmentation and Attractor Identification** Given the demonstration set $\Xi$ and AP regions related to the set of $\mathcal{M}$ modes we can automatically segment the trajectories into $\mathcal{M}$ clusters and corresponding attractor set $X^*$. For more details, refer to Appendix C.
**Ensuring Goal Reachability with Learned DS Mode Policies** While any BC variant with a stability guarantee can satisfy reachability (see Sec. 2), we focus on the G.A.S. DS formulation and learning approach defined in Section 4.3 that ensures every $x \in \mathbb{R}^n$ is guaranteed to reach $x_i^*$. By placing $x_i^*$ within the boundary set of $\delta_j$ for a mode $\sigma_j$, we ensure mode $\sigma_j$ is reachable from every $s$ in mode $\sigma_i$. Note $f(x)$ cannot model sensor dynamics in $\alpha$. Yet, we employ mode abstraction to reduce the imitation of a system state trajectory in $s$—which includes the evolution of both the robot and sensor state—to just a robot state trajectory in $x$.

### 5.2 Online Learning Phase
**Iterative Mode Boundary Estimation via Invariance Failures** As shown in Fig. 2, DS can suffer from *invariance* failures in regions without data coverage. Instead of querying humans for more data in those regions [20], we leverage sparse events of mode exits detected by sensors to estimate the unknown mode boundary. Specifically, for each invariance failure, we construct a cut that separates the failure state, $x^{T_f}$, from the mode-entry state, $x^0$, the last in-mode state, $x^{T_f-1}$, and the mode attractor, $x^*$. We ensure this separation constraint with a quadratically constrained quadratic program (QCQP) that searches for the normal direction (pointing away from the mode) of a hyperplane that passes through $x^{T_f-1}$ such that the plane's distance to $x^*$ is minimized. The intersection of half-spaces cut by the hyper-planes inner-approximates a convex mode boundary, as seen in Fig. 4. Adding cuts yields better boundary estimation, but is not necessary unless the original vector field flows out of the mode around those cuts. For more details, refer to Appendix E.3.
**Ensuring Mode Invariance by Modulating DS** We treat each cut as a collision boundary that deflects the DS flows following the approach in [44, 53]. In our problem setting the mode boundary is analogous to a workspace enclosure rather than a random task-space object. Let existing cuts form an implicit function, $\Gamma(x) : \mathbb{R}^n \to \mathbb{R}^+$, denote the estimated interior with $\Gamma(x) < 1$, $\Gamma(x) = 1$ the boundary and $\Gamma(x) > 1$ the exterior of a mode. $0 < \Gamma(x) < \infty$ monotonically increases as $x$ moves away from a reference point $x^r$ inside the mode. For $x$ outside the cuts, or inside but moving away from the cuts, we leave $f(x)$ unchanged; otherwise, we modulate $f(x)$ to not collide with any cuts as $\dot{x} = M(x)f(x)$ by constructing a modulation matrix $M(x)$ through eigenvalue decomposition:

$$\begin{cases} M(x) = E(x)D(x)E(x)^{-1}, \;\; E(x) = [\mathbf{r}(x)\; \mathbf{e}_1(x) \; ... \; \mathbf{e}_{d-1}(x)], \;\; \mathbf{r}(x) = \frac{x-x^r}{\|x-x^r\|} \\ D(x) = \mathbf{diag}(\lambda_r(x), \lambda_{e_1}(x), ..., \lambda_{e_{d-1}}(x)), \;\; \lambda_r(x) = 1 - \Gamma(x), \;\; \lambda_e(x) = 1 \end{cases} \quad (7)$$

The full-rank basis $E(x)$ consists of a reference direction $\mathbf{r}(x)$ stemming from $x^r$ toward $x$, and $d-1$ directions spanning the hyperplane orthogonal to $\nabla\Gamma(x)$, which in this case is the closest cut to $x$. In other words, all directions $\mathbf{e}_1(x)...\mathbf{e}_{d-1}(x)$ are tangent to the closest cut, except $\mathbf{r}(x)$. By modulating only the diagonal component, $\lambda_r(x)$, with $\Gamma(x)$, we have $\lambda_r(x) \to 0$ as $x$ approaches the closest cut, effectively zeroing out the velocity penetrating the cut while preserving velocity tangent to the cut. Consequently, as long as there are cuts bounding the mode, the modulated DS will not experience invariance failures where the original DS would. Notice this modulation strategy is not limited to DS, and can apply to any state-based BC methods to achieve mode invariance.

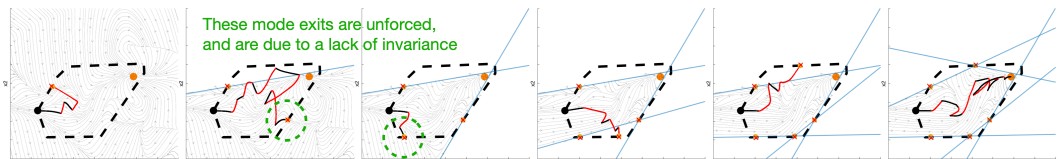

Figure 4: An illustration of iterative estimation of mode boundary with cutting planes. A system enters a mode with an unknown boundary (dashed line) at the black circle, and is attracted to the goal at the orange circle. Its trajectory in black shows the original policy rollout, and its trajectory in red is driven by perturbations. After the system exits the mode and before it eventually re-enters the same mode through replanning, a cut is placed at the last in-mode state (yellow circle) to bound the mode from the failure state (red cross). When the system is inside the cuts, it experiences modulated DS that never leaves the mode (flows entering the mode are not modulated); when the system is outside the cuts but inside the mode, it follows the original DS. Note only mode exits in black are invariance failures in need of modulation (green circles); mode exits in red are driven by perturbations to illustrate that more cuts lead to better boundary approximation.

## 6 Proof

Next, we prove LTL-DS produces a continuous trajectory that satisfies a LTL specification. We start with assumptions and end with theorems. Detailed proofs are provided in Appendix A.

**Assumption 1.** *All modes are convex.*

This assumption leads to the existence of at least one cut—i.e., the supporting plane [54], which can separate a failure state on the boundary from any states within the mode. A corollary is that a boundary shared by two modes, which we call a guard surface, $G_{ij} = \delta_i \cap \delta_j$, is also convex. Since all transitions out of a mode observed during demonstrations reside on the mode boundary, their average location—which we use as the attractor for the mode—will also be on the boundary.

**Assumption 2.** *There are a finite number of externally exerted motion- and task-level perturbations of arbitrary magnitude.*

Given zero perturbation, all BC methods should succeed in any task replay, as the policy rollout will always be in distribution. If there are infinitely many arbitrary perturbations, no BC methods will be able to reach a goal. Here, we study the setting between these extremes—where there are a finite number of external perturbations causing unexpected mode exits.

**Assumption 3.** *Every unexpected mode transition only results in sensor states that have been seen in the demonstrations.*

While demonstrations of all valid mode transitions are not required, they must minimally cover all possible modes. If the system encounters a completely new sensor state during online interaction, it is reasonable to assume that no BC methods could recover from the mode unless more information about the environment were provided.

**Theorem 1.** *(Key Contribution 1) A nonlinear DS defined by Eq. 4, learned from demonstrations, and modulated by cutting planes as described in Section 5.2 with the reference point $x^r$ set at the attractor $x^*$, will never penetrate the cuts and is G.A.S. at $x^*$.* ***Proof:*** *See Appendix A.* □

**Theorem 2.** *(Key Contribution 2) The continuous trace of system states generated by LTL-DS defined in Eq. 6 satisfies any LTL specification $\phi$ under Asm. 1, 2, and 3.* ***Proof:*** *See Appendix A.* □

## 7 Experiments

### 7.1 Single-Mode Invariance and Reachability

We show quantitatively both reachability and invariance are necessary for task success. We compare DS and a NN-based BC policy (denoted as BC) to represent policies with and without a stability guarantee respectively. Fig. 5 shows that policy rollouts start to fail (turn red) as increasingly larger perturbations are applied to the starting states; however, DS only suffers from invariance failures, while BC suffers from both invariance and reachability failures (due to diverging flows and spurious attractors). Fig. 5 (right) shows that all flows are bounded within the mode for both DS and BC after two cuts. In the case of DS, flows originally leaving the mode are now redirected to the attractor by the cuts; in the case of BC, while no flows leave the mode after modulation, spurious attractors are created, leading to reachability failures. This is a counterfactual illustration of Thm. 1, that policies without a stability guarantee are not G.A.S. after modulation. Fig. 6 verifies this claim quantitatively and we empirically demonstrate that a stable policy requires only four modulation cuts to achieve a perfect success rate—which an unstable policy cannot be modulated to achieve.

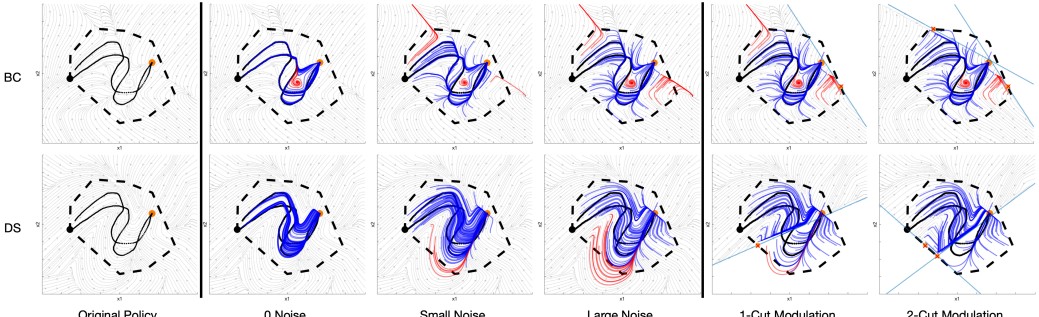

Figure 5: Policy rollouts from different starting states for a randomly generated convex mode. The top row shows BC results, and the bottom row depicts DS results. The left column visualizes the original policies learned from two demonstrations (black trajectories) reaching the orange attractor. The middle columns add different levels of Gaussian noise to the initial states sampled from the demonstration distribution. Blue trajectories successfully reach the attractor, while red trajectories fail due to either invariance failures or reachability failures. (Note that these failures only occur at locations without data coverage.) The right columns show that cutting planes (blue lines) separate failures (red crosses) from last-visited in-mode states (yellow circles), and consequently can modulate policies to be mode-invariant. Applying cutting planes to BC policies without a stability guarantee cannot correct reachability failures within the mode. More results are in Appendix E

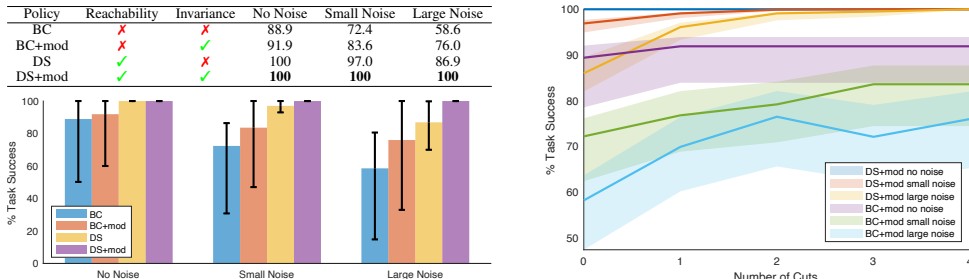

Figure 6: **(left)** The success rate (%) of a single-mode reaching task. As we began to sample out of distribution by adding more noise to the demonstrated states, the BC's success rate degraded more rapidly than the DS'. After modulation, DS (+mod) maintained a success guarantee, which BC (+mod) fell short of due to the base policy's lack of a stability guarantee. **(right)** Empirical evidence that single-mode invariance requires only a finite number of cuts for a base policy with a stability guarantee. Regardless of the noise level, DS achieves a 100% success rate after four cuts, while BC struggles to improve performance with additional cuts. Thick lines represent mean statistics; shaded regions the interquartile range. More details are provided in Appendix E.

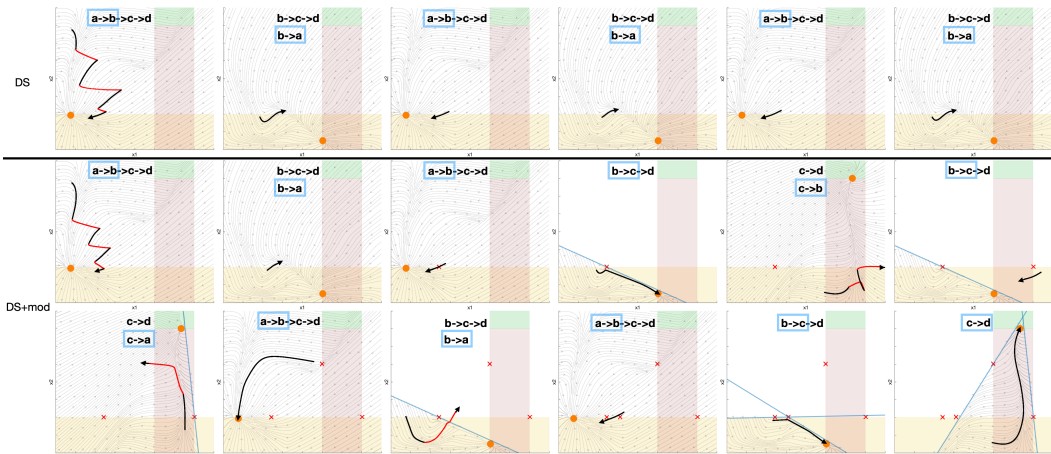

Figure 7: Rollouts of a multi-step scooping task under perturbations. The first row indicates that DS policies sequenced by an automaton but without boundary estimation can lead to looping; the second and third rows show that modulation can prevent looping and eventually allow the system to reach the goal mode despite repeated perturbations. We depict the mode sequence planned by the automaton at the top of each sub-figure, and blue bounding boxes indicate the current mode transitions actually being detected. Black and red trajectories signify original and perturbed policy.

### 7.2 Multi-Modal Reactivity and Generalization to New Tasks

We now empirically demonstrate that a reactive discrete plan alone is insufficient to guarantee task success without mode invariance for tasks with multiple modes. Consider the multi-modal soup-scooping task introduced in Fig. 2. Formally, we define three environment APs, $r, s, t$, sensing the spoon is in contact with the soup, has soup on it, and has arrived at a target location respectively. Given successful demonstrations, sensors will record discrete transitions $(\neg r \wedge \neg s \wedge \neg t) \Rightarrow (r \wedge \neg s \wedge \neg t) \Rightarrow (\neg r \wedge s \wedge \neg t) \Rightarrow (\neg r \wedge \neg s \wedge t)$, from which four unique sensor states are identified. We label each sensor state as a mode with robot AP $a$ (reaching) $\Rightarrow b$ (scooping) $\Rightarrow c$ (transporting) $\Rightarrow d$ (done). Invariance of mode $b$ enforces contact with soup during scooping, and invariance of mode $c$ constrains the spoon's orientation in order to avoid spilling. We follow the TLP convention to assume LTL formulas are provided by domain experts (although they can also be learned from demonstrations [51, 55].) The specific LTL for the soup-scooping task is detailed in Appendix F, and can be converted into a task automaton as shown in Fig. 3.

One might assume the automaton is sufficient to guarantee task success without modulation, as it only needs to replan a finite number of times assuming a finite number of perturbations; however, not enforcing mode invariance can lead to looping at the discrete level, and ultimately renders the goal unreachable, as depicted in the top row of Fig. 7. In contrast, looping is prevented when modulation is enabled, as the system experiences each invariance failure only once. We found $50\%$ of the learned policies without modulation could get stuck in looping after a finite number of perturbations, while all experiments with modulation succeeded in the task replay.

**Robot Experiments** First, we implemented the soup-scooping task on a Franka Emika robot arm as shown in Fig. 1. We show in videos on our website that (1) DS allowed our system to compliantly react to motion-level perturbations while ensuring system stability; (2) LTL allowed our system to replan in order to recover from task-level perturbations; and (3) our modulation ensured the robot learned from previous invariance failures in order to avoid repeating them. To test robustness against unbiased perturbations, we collected 25 trials from 5 humans as seen in in Appendix H. All trials succeed eventually in videos. We did not cherry-pick these results, and the empirical 100% success rate further corroborates our theoretic success guarantee. Second, we implemented an inspection task as a permanent interactive exhibition at MIT museum, with details documented in Appendix I. Lastly, we show a third color tracing task testing different automaton structures on our website.

**Generalization** Once a DS is learned we can generalize to a new task sharing the same set of modes observed in demonstrations given a new LTL formula. Consider another multi-step task of adding chicken and broccoli to a pot. Different humans might give demonstrations with different modal structures (e.g., adding chicken vs adding broccoli first). LTL-DS could learn individual DS which can be flexibly combined to solve new tasks with new task automatons. To get these different task automatons, a human just needs to edit the $\phi_t^s$ portion of the LTL formulas differently. We provide further details of this analysis in Appendix G.

## 8 Limitations

Our approach of integrating logic formulas into LfD requires defining an appropriate abstraction for a task. For example, allocating a sensor to detect contact events requires domain knowledge. Our work is based on the assumption that for well-defined tasks (e.g., assembly tasks in factory settings), domain expertise in the form of a logic formula is a cheaper knowledge source than collecting hundreds of motion trajectories to avoid covariate shift (we use up to 5 demonstrations in all experiments). However, even when abstractions for a task are given by an oracle, a LfD method without either the invariance or the reachability property will not have a formal guarantee of successful task replay, which is this work's focus. In future work, we will learn such abstractions directly from sensor streams such as videos so that our approach gains more autonomy without losing reactivity.

## 9 Conclusion

In this paper, we formally introduce the problem of *temporal logic imitation* as imitating continuous motions that satisfy a LTL specification. We identify the fact that learned policies do not necessarily satisfy the bisimulation criteria as the main challenge of applying LfD methods to multi-step tasks. To address this issue, we propose a DS-based approach that can iteratively estimate mode boundaries to ensure invariance and reachability. Combining the task-level reactivity of LTL and the motion-level reactivity of DS, we arrive at an imitation learning system able to robustly perform various multi-step tasks under arbitrary perturbations given only a small number of demonstrations, and demonstrate our system's practicality on a real Franka robot.

## Acknowledgments

We would like to thank Jon DeCastro, Chuchu Fan, Terry Suh, Rachel Holladay, Rohan Chitnis, Tom Silver, Yilun Zhou, Naomi Schurr, and Yuanzhen Pan for their invaluable advice and generous help.

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
