# OpenReview forum: "Temporal Logic Imitation: Learning Plan-Satisficing Motion Policies from Demonstrations"
_robot-learning.org/CoRL/2022/Conference — CoRL 2022 Oral_

### Official Review · Reviewer_p7Tc · 2022-07-06

**Originality:** Good
**Technical Quality:** Very Good
**Clarity Of Presentation:** Good
**Impact:** 2

**Recommendation:**

Strong Accept: I recommend accepting the paper and will argue for my recommendation even if other reviewers hold a different opinion.

**Summary:**

This paper presents a temporal logic imitation approach which generates policies that satisfy a given linear temporal logic (LTL) specification.
Classical learning from demonstration approaches (LfD) can often not ensure task satisfaction in the presence of perturbations.
To address this issue, the presented approach proposes a mode-based approach to satisfy the task while optimizing controllers that ensure stability.
The task is decomposed into a set of modes, e.g., moving a spoon to a soup, scooping the soup, and moving to the goal pose.
Local controllers within each mode ensure stability in the presence of perturbations.
The paper shows that the presented approach satisfies invariance and reachability for the modes.
The approach has been evaluated in simulation and on a real robot in a soup-scooping task.
Results indicate that the approach significantly improves performance under various perturbations.

**Issues:**

* Temporal logic imitation: How can one create specifications that can be used in the approach?
Which conditions need to be fulfilled to specify transitions between modes?
* Assumptions: which assumptions do the authors make in their approach? Which tasks can this approach be applied to?
* Experiments: More details on the task, the specification, and the evaluation criteria.
* Approach: More details on choosing the attractors.
* Limitations: What are limitations of the approach? When does the approach fail? Can it be applied to all tasks?

**Quality Of The Limitations Section:**

Limitations are not well addressed

**Reviewer Expertise:**

4: The reviewer is confident but not absolutely certain that the evaluation is correct

**Robotics Focus:**

Relevant but unlikely to deploy to hardware in near future

**Strengths And Weaknesses:**

Robustly performing tasks under various perturbations is challenging.
This paper presents an interesting and promising control-theoretic approach to address this challenge.
The approach builds upon sophisticated control theory.
The proposed task abstraction into modes is compelling and helps to decompose the task.
Constructing stable controllers for each mode to create plans is a key feature of this work.
The authors also provide various proofs and details on how to create the controllers and the convex operating regions in each mode.
The approach can work with various user-defined LTL formulas that specify how modes can be entered and how the system should transition from one mode to another.
The authors provide detailed explanations on the mode and how to construct the controllers.
The approach has been evaluated in a small simulation with various controlled perturbations and on a real robot which is great to see.
The submitted videos help to understand the approach better and showcase various aspects of the approach.

Although the paper is interesting, its scope is not really fitting the learning theme of the conference.
The paper claims to address learning from demonstration.
Yet, the approach focuses on control-theoretic aspects (both main contributions are Theorem 1 and 2) and learning is left out.
The only reference to learning is on learning the linearized dynamics through Bayesian non-parametric GMMs.
The demonstrations as well as the LTL specifications are assumed to be provided to the approach.
It would have been great to see more integration into learning from demonstration frameworks.
The control theoretic details are great, but the paper does not really fit the scope of CoRL.
Moreover, the paper claims to focus on LTL specifications and it introduces the necessary foundations.
Except for one experiment, the paper does not talk about LTL anymore.
Instead, specifications are assumed to be provided by experts.
How can one create specifications that can be used in the approach?
Which conditions need to be fulfilled to specify transitions between modes?
Which limitations does the approach have?
In its current form, the paper does not provide enough details to justify the general claims of temporal logic imitation (the paper focuses on modes and constructing controllers).
Considering the applicability, the paper has focuses on a rather simple task for which the mode decomposition is straightforward.
How does the approach translate to other tasks?
What if the task contains time-variant elements, such as dynamic objects?
In the control sections, the paper needs to provide more details.
For instance, how are attractors chosen?
What kind of assumptions are made on the system/sensors?
What happens if the system's dynamics are highly non-linear and the approximation in the paper leads to large linearization errors?
The paper also misses to explain the motivation behind many modeling steps, e.g., why are the controllers chosen as presented?
Lastly, the experiment section is very short, and it is not possible for a reader to follow the details and contributions of the experiments.
It would be nice if the paper can properly introduce the considered environment, system, and task.
The limitations sections can also be improved by not just stating straightforward future work, but discussing the limitations of the approach and its applicability to a wide range of tasks.

Considering the paper's style, it is mostly well written.
Yet, the control-theoretic sections are very dense due to the page limit.
The authors could remove the unused LTL definitions to gain space and add more explanations.
Moreover, the figures are very small, and it would be nice if the authors could add labels to them for easier reference.
The numbering of equations 3 and 4 can also be improved since it is not directly clear where Eq. 3 is.

**Summary Of Recommendation:**

In summary, the paper presents an interesting control-theoretic approach that shows promising results.
However, the paper has no real focus on learning and seems out of scope for CoRL.
The integration with learning from demonstrations is not really described in the paper.
Moreover, the approach does not really demonstrate temporal logic imitation.
The specifications are assumed to be provided and the paper does not explain how to create specifications for the use in the approach.
THe paper also does not discuss the temporal logic aspects in detail.
The experimental section also needs to be revised to properly demonstrate the approach and claims in various environments and tasks.

---

> ### Author Response · Authors · 2022-08-22
> **Response to reviewer p7Tc (part 1)**
>
> Dear reviewer p7Tc:
>
> We really appreciate your detailed feedback and positive evaluation of our originality, technical quality, and clarity of presentation despite the rejection. We hope the following response can clarify any misunderstanding and assuage your concerns:
>
> **[Scope not fitting]** The primary concern of the reviewer is that our work is not a learning algorithm. This impression might have come from our previous structure of the text where we leave many details in the appendix. **We have since restructured the text to highlight the learning components in our approach in the newly submitted section 3, section 5, and appendix H.** In particular, we explain that LTL-DS is a form of imitation learning augmentation defined as “Teacher-Student Interaction during Behavioral Cloning” in [1], and can be categorized alongside the well-established LfD algorithm DAGGER [2]. While a pure control strategy without any learning could solve problems in TLP (explained in appendix H), it is not sufficient to solve problems in our TLI setting which inherit challenges unique to LfD. Specifically, LTL-DS requires offline demonstration data to learn the mode policies and online interaction data to learn the mode boundaries on a physical robot **(paper outcome)**. The control theoretic formulation of the mode policy and the task imitation policy are necessary to certify an LfD method w.r.t. LTL task specifications **(paper intent).** Hence our approach is both data-driven and control-theoretic with close ties to learning in terms of intent and outcome (CoRL in-scope criteria). We hope this additional clarification helps assuage the reviewer's concern.
>
>
> **[How to create LTL specification]** We would like to clarify that TLI does not tackle the problem of learning LTL specifications from demonstrations, which itself is a well-studied field as mentioned in Appendix H "Relation of TLI to Prior Work". To specify LTL in practice, we have explained in the original text (1) how to create the scooping task LTL in Appendix C (line 512), (2) how to automatically construct formulas under mild assumptions in Appendix C (line 519), and (3) how to create the generalization results LTL in Appendix D (line 547). Apart from the GR(1) formula template detailed in these Appendix sections, there are no additional conditions needed to specify mode transitions.
>
> **[What assumptions/limitations do authors make/concede?]** In terms of assumptions, we have dedicated (1) the TLI formulation section 3 (both the original and newly submitted text) to explain the assumptions on the systems/sensors, (2) the proof section 5 (original text) to explain three assumptions we are making for the proof to go through. Given those three assumptions, our approach can apply to any arbitrarily complex task in theory as long as the task can be represented by a sequence of convex modes and the task can be demonstrated successfully by a human. In practice, the implementation on the hardware is limited by the availability of good sensors to detect mode transitions accurately as mentioned in the limitation section 7 (original text). More specifically, if the assumption to have good sensors is violated (e.g., if lighting affects the color detection accuracy of cameras), the theoretical success guarantee won't translate into a hardware experiment success guarantee. However, in this case, the failure is not due to our algorithm but rather the hardware limitations that will equally plague any other methods mentioned in our related works section. In terms of time-varying aspects of a task, DS can handle dynamic objects as long as the DS attractor is moving along (grounded on) the object [3], which can be tracked by real-time sensors such as optitrack. We appreciate and welcome the reviewer's questions regarding our limitations, and we will incorporate the above explanations into an expanded limitation section in the revised draft.

---

> > ### Author Response · Authors · 2022-08-22
> > **Response to reviewer p7Tc (part 2)**
> >
> > **[More details on the controller]** The reviewer asks how the attractor is chosen. The attractor for each mode transition is chosen as the average of all locations of such mode transitions seen in the demonstrations. **Please refer to the newly submitted appendix I for details on attractor identification.** The reviewer is also concerned the linearization in DS might bottleneck the representation of non-linear motions. First, the DS variant we are using is a SoTA LfD method learning ultra-nonlinear motions by a mixture of linear models as seen in [4]. Even if the learning of nonlinear motions is not perfect, the evaluation criteria in our task is not how accurate the robot can reproduce the demonstrations, but rather if the robot can be guaranteed to reach the goal mode despite arbitrary perturbations. **If there were learning errors that were large enough to cause unexpected mode exits, the reactivity of LTL would replan and recover from those mistakes, which is precisely the strength of this work.** Lastly, the reviewer asks for the motivation to use DS as the controller. This is explained in the "perturbations" part of the newly submitted section 3 and section 4. In short, having motion-level robustness requires a motion policy to be stable, which DS satisfies by construction. In contrast, in section 6.1 (line 233 of the original text) we show an LfD method without stability (BC) after modulation will not attain 100% task success. Note our approach is not limited to using DS, which can be replaced by any other stable motion policies mentioned in the newly submitted section 5.1.
> >
> >
> > **[More details on the experiments]** The reviewer is concerned the experiment section is too short to highlight the contribution of the experiments. We'd first like to clarify that experiments in the paper are not the main contribution of this work. As our problem formulation has not been studied before (as explained in Appendix H "Relation of TLI to Prior Work"), we hope to establish the theory first in this paper. We had only intended the empirical experiments in the paper to motivate and illustrate our **main theoretical contribution to the LfD community, i.e. a novel problem formulation and a formal proof to certify LfD methods.** We would in follow-up work conduct a more extensive empirical study of the proposed approach on a task suite such as [5]. That being said, we do have three categories of simulation tasks and a robot experiment robust to perturbations. While the experiment section in the main text is short due to the page limit, we have extensive experiment details documented in Appendix B, Appendix C, Appendix D, and Appendix E. We invite the reviewer to take a look at them. If more details are missing from the experiments, we are glad to incorporate them. Lastly, we are currently working on a second robot experiment involving a completely different task, sensor states, and LTL formula. We will include it in the appendix section in the next couple of days. Thanks for your patience!
> >
> > We hope our response has adequately addressed your concerns such that you would judge this work more favorably. Please let us know if you have any additional specific questions or need more elaborations from us on any of the existing ones. We would love to work together to make this a better paper. Thank you!
> >
> > [1] Osa, et al. An Algorithmic Perspective on Imitation Learning
> >
> > [2] Ross, et al. A reduction of imitation learning and structured prediction to no-regret online learning.
> >
> > [3] Salehian, et al. A Dynamical System Approach for Softly Catching a Flying Object: Theory and Experiment
> >
> > [4] Figueroa, Billard. A Physically-Consistent Bayesian Non-Parametric Mixture Model for Dynamical System Learning
> >
> > [5] James, et al. RLBench: The Robot Learning Benchmark & Learning Environment

---

> > > ### Author Response · Authors · 2022-08-25
> > > **Response to reviewer p7Tc: additional evaluations on the real robot**
> > >
> > > **Comment:**
> > >
> > > **[New results: additional human experiments]** Since external perturbations are an integral part of our task complexity, we recruited five human subjects without prior knowledge of our LTL-DS system to perturb the robot scooping setup. Each subject is given five trials of perturbations. In total, we collected 25 trials, each of which is seen as an unbiased i.i.d. source of perturbations. In our newly submitted video (which we also upload to the project page https://sites.google.com/view/ltl-ds), **we show all 25 trials succeed eventually. We did not cherry-pick the results and we shoot all videos in one go.** This empirical 100% success rate further corroborates our theoretic success guarantee. Interestingly, common perturbation patterns (we annotate with the same colored text) emerge from different participants. Specifically, we see adversarial perturbations where humans fight against the robot and collaborative perturbations where humans help the robot to achieve the goal of transferring at least one bead from one bowl to the other. In the case of adversarial perturbations, DS reacts and LTL replans. In the case of collaborative perturbations, DS is compliant and allows humans to also guide the motion. In the case where humans are not perturbing yet the robot makes a mistake (e.g. during scooping), LTL replans the scooping DS until the robot enters the transferring mode successfully. The fact that we don't need to hard code different rules to handle invariance failures caused by perturbations and the robot's own execution failures in the absence of perturbations highlights the strength of our LTL-powered sensor-based task reactivity. We will continue to present more robot experiments in the coming days.
> > >
> > > **Zip File:**
> > >
> > > /attachment/e721d7c590734918445e175bf6513682469d1731.zip

---

> > > > ### Comment · Reviewer_p7Tc · 2022-08-25
> > > > **Response to authors on new experiment and clarifications**
> > > >
> > > > Dear authors,
> > > > Thank you for your detailed answers, clarifications on potential misunderstandings, and your additional work. The new descriptions and the comparison with respect existing problems help to understand and categorize the studied problem much better.
> > > > The clarifications on the paper's assumptions as well as current limitations and future work will make the paper much stronger. Since the proposed approach provides guarantees, it is now easier for readers to understand the approach's capabilities, e.g., with the added explanations and comparison of the controller part.
> > > > Lastly, the new experiments with humans as unbiased perturbation sources are very helpful to stress test the approach and demonstrate its maturity/ rigorous guarantees. The video of the experiments is very interesting and the discussion on the perturbation patterns insightful. Designing and performing additional experiments in such a short amount of time is not easy and the new results are very much appreciated.  Considering all the changes and additional work, I believe that the paper's quality has much improved!

---

> > > > > ### Author Response · Authors · 2022-08-28
> > > > > **Final response to reviewer p7Tc**
> > > > >
> > > > > **Comment:**
> > > > >
> > > > > Dear reviewer,
> > > > >
> > > > > We are attaching three additional robot experiment videos to address your concern about too few tasks. Experiment 1 is explained in our previous response to all reviewers. Experiment 2 is documented in the new Appendix J. Experiment 3 is testing different automaton structure, and we will include more details in the camera-ready version should the paper gets accepted. Thank you for your time and consideration!
> > > > >
> > > > > **Zip File:**
> > > > >
> > > > > /attachment/bb2d5281b3a37760ca36e7ad7bcb182dde16c2d2.zip

---

### Official Review · Reviewer_L7Wt · 2022-07-14

**Originality:** Very Good
**Technical Quality:** Excellent
**Clarity Of Presentation:** Excellent
**Impact:** 4

**Recommendation:**

Strong Accept: I recommend accepting the paper and will argue for my recommendation even if other reviewers hold a different opinion.

**Summary:**

Current learning from demonstration (LfD) techniques, do not guarantee success in the presence of significant perturbations, and rarely have the ability to recover from task failures.  In this paper, the authors frame this deficiency as a failure to satisfy the high-level task specification which is implicitly present in the given demonstrations.

By leveraging task "modes" (the sets of sensor states which correspond to relevant components of the task) in combination with a LTL-automaton and dynamical system, their method is able to satisfy the task in the presence of arbitrary perturbation, and even recover from perturbations which cause previous steps of a task-plan to be undone.

In contrast with work in temporal logic planning, these "modes" are not given, but learned from the data with a quadratically constrained quadratic program to learn the boundary cutting planes of each mode.

Their experiments demonstrate their method with a soup-scooping task on a Franka robot arm.

**Issues:**

Paper is essentially good in its current form. I was left wondering about the limitations and expressive potential of the author's chosen constraint language (the GR(1) fragment of LTL). How broadly does this apply to robotic missions in general as compared to say Signal Temporal Logic, or Probabilistic Signal Temporal Logic? Perhaps this could get a brief mention in the limitations section.

**Quality Of The Limitations Section:**

Limitations are addressed clearly

**Reviewer Expertise:**

4: The reviewer is confident but not absolutely certain that the evaluation is correct

**Robotics Focus:**

Sufficient demonstration on hardware

**Strengths And Weaknesses:**

# Strengths

The presentation is clear, and the formalizations are clean & understandable. The figures do a good job providing intuitive explanations of algorithm operation, and how it contrasts with traditional imitation learning methods.

The work is well situated --- alternative solutions to their problem are addressed and discussed (e.g., framing the problem as covariate shift and simply asking for more human demonstrations). The related work covers the relevant literature on temporal logic motion planning, imitation learning, and subgoal/long-horizon manipulation.

The contribution itself is strong, representing progress in augmenting LfD frameworks to incorporate task specification knowledge in a way that both robust to failures and aids recovery. In addition to empirical evidence for the value of their technique, the theoretic proofs clearly outline the assumptions under which their method can manage arbitrary perturbations.

The experiments were implemented on a real panda robot, and fair comparisons are made to NN-Behaviour Cloning policies to demonstrate the advantages of their method.

# Weaknesses

The weaknesses I can identify mostly come from a desire to see this work extended.

For one, while the soup scooping example is cleanly demonstrates their idea, I would have liked to see it applied to a range of varied tasks with expressive modes.

Additionally, while the modes of the task are learned, the high-level sensor abstractions themselves used to learn these modes are given. It would be interesting to see if this approach could be applied to high-dimensional data-driven perception (e.g., streaming video pushed through a perception network). I note that this is mentioned by the authors in their own limitation section.

Third, their use of GR(1) LTL as a constraint language inherently limits the types of cyber-physical problem that can be addressed with this approach. For plans involving continuous signals, timing constraints, or looser probabilistic guarantees, a more expressive language would likely be needed. I am unsure if transferring these techniques to something like P-STL would be a non-trivial endeavor.

**Summary Of Recommendation:**

I feel confident in recommending this paper for publication --- the presentation is clear, the techniques novel, it has a strong formal grounding, and is well placed within the existing LfD literature.

---

> ### Author Response · Authors · 2022-08-22
> **Response to reviewer L7Wt**
>
> Dear reviewer L7Wt:
>
> Thanks for your detailed review and encouraging comments! Please find our response below:
>
> **[Expressivity of LTL]** We use the GR(1) fragment due to its ease of planning following [1]. In general, our method could translate to any automaton representation but is time efficient if the LTL specification is in GR(1) format. The reviewer is correct that LTL cannot handle all types of robotics missions, and it is less expressive than STL and PSTL. Unfortunately, we rely on an automaton representation for our approach, and we cannot compile STL and PSTL specifications into an automaton format. Thus the extension of our work to STL and PSTL will be non-trivial. However, we can represent STL, PSTL, and LTL in a mixed integer constraint satisfaction problem, and we would explore the integration of mixed integer constraint satisfaction solvers with our present approach in future work. We thank the reviewer for the suggestion to consider more expressive task specifications.
>
> **[More tasks would be better]** In the paper, we currently have three categories of tasks: (1) Single-mode reaching tasks, (2) Multi-modal scooping tasks (3) Multi-modal generalization tasks (in the appendix due to page limit). Admittedly, our contribution is not an extensive empirical evaluation of any particular LfD algorithm on a suite of tasks such as [2]. As our problem formulation has not been studied before and the theory is missing, we had only intended the empirical experiments in the paper to motivate and illustrate our theoretical contribution to the LfD community, i.e. a formal proof to certify LfD methods. That being said, we are currently working on some additional robot demos per the reviewer's request and will include them in the appendix section in the next couple of days. Thanks for your patience!
>
> [1] Kress-Gazit, et al. Temporal-Logic-Based Reactive Mission and Motion Planning
>
> [2] James, et al. RLBench: The Robot Learning Benchmark & Learning Environment

---

> > ### Author Response · Authors · 2022-08-28
> > **Final response to reviewer L7Wt**
> >
> > **Comment:**
> >
> > Dear reviewer,
> >
> > We are attaching three additional robot experiment videos to address your concern about too few tasks. Experiment 1 is explained in our previous response to all reviewers.  Experiment 2 is documented in the new Appendix J. Experiment 3 is testing different automaton structure, and we will include more details in the camera-ready version should the paper gets accepted. Thank you for your time and consideration!
> >
> >
> > **Zip File:**
> >
> > /attachment/2166f53d24e926bbc34efbc92ec7ed157f4dab7e.zip

---

### Official Review · Reviewer_QTy6 · 2022-07-30

**Originality:** Good
**Technical Quality:** Very Good
**Clarity Of Presentation:** Good
**Impact:** 3

**Recommendation:**

Weak Accept: I recommend accepting the paper, but will not argue for my recommendation if the majority of other reviewers have a different opinion.

**Summary:**

The paper presents an approach to train sets of controllers to imitate demonstrations. The demonstrations can be segmented into sequences of steps separated by sensory events, leading to a representation of the tasks in the form of hybrid automata with sequences of sensory "modes". The method combines synergistically prior work on dynamical systems to obtain convergent controllers for each of the segments/steps, and prior work on temporal-logic synthesis to map the demonstrations into hybrid automata, to obtain convergent policies both in the motion level and in the task level (sequences of modes). The method is tested on synthetic data and with a proof of concept on a real robot.


**Issues:**

Mentioned in the weaknesses:
- Make clear what is prior work and what is novel contribution, possibly with a "Preliminaries" section
- Additional tasks and evaluation conditions on the real robot

**Quality Of The Limitations Section:**

Additional details required

**Reviewer Expertise:**

4: The reviewer is confident but not absolutely certain that the evaluation is correct

**Robotics Focus:**

Sufficient demonstration on hardware

**Strengths And Weaknesses:**

Strengths:
- The paper is well written, although very tightly packed
- The community is in need of methods that move away from “just” action imitation into some forms of imitation with better abstractions and objectives
- The combination of control with convergence warranties with a hybrid automata is novel and interesting for the community.
- It includes a demonstration of the approach on a real-robot

Weaknesses:
- The experimental evaluation is rather limited to understand the strengths and limitations. Many of the concepts are “evaluated” in simple synthetic 2D point-agent domains that, while illustrative, are more a proof of concept. There is only one task with full-robotic complexity. This makes it hard to understand the weaknesses of the proposed approach. For example, how well would it scale to more complex automata? What is the complexity of finding and perceiving the sensor events to switch between modes (partially mentioned in the limitations)? How stable is the boundary in the switch between states of the automata (usually, the problematic parts)?
- There are a lot of concepts that are inherited from the theory of dynamic systems and temporal logic and that were presented before. While I appreciate the effort to make the paper self-contained with these explanations, the paper does not draw a clear line on what is novel and what is prior work applied here. A suggestion is to create a section “Preliminaries” to explain the used prior concepts that would make very clear what is new.
- As far as I understand, the videos/experiments with the robot show the same person that created the solution (and knows the limitations) perturbing the agent. Some of the perturbations seem to be studied and I don't know if they are similar to the ones applied to the (failing) baselines. While this is good as a proof of concept, it would be great to perform some experiments with unbiased participants.



**Summary Of Recommendation:**

It is a good contribution, well written and well motivated. The paper is technically competent, although it is hard to understand what is novel contribution and what is prior work in dynamical systems and temporal-logic. The experiments are illustrative but additional robot experiments (more tasks, unbiased perturbations) would help understand the strengths and limits of the presented approach.

---

> ### Author Response · Authors · 2022-08-22
> **Response to reviewer QTy6 (part 1)**
>
> Dear reviewer QTy6,
>
> We appreciate your detailed review and questions! Please find our response below:
>
> **[Distinction from prior work]** We uploaded a supplementary file including (1) a re-write of sections 3-5 into preliminaries and additional details on our approach for clarification (If accepted we will incorporate these changes to the main text), and (2) a new appendix H “Relation of TLI to Prior Work”, where we explain how our work and prior work fit together.
>
> **[Scalability to more complex automata]** The reviewer is concerned that our method LTL-DS might not scale well to more complex automata given many experiments are demonstrated only on 2D toy tasks. First, we test our algorithm on various chicken and broccoli tasks with non-trivial automaton structures in appendix D. Second, we assume LTL formulas are given, and our proof certifies that LTL-DS can be modulated to satisfy any arbitrarily complex LTL and thus scale to their corresponding arbitrarily complex automata. Third, neither our modulation strategy nor our proof makes an assumption on the dimensionality of the state space and thus is not limited to 2D tasks.  Admittedly, the scalability w.r.t the complexity of task automata in this work is a theoretical result, and thus is subjected to the limitations of hardware implementations such as sensor noise (which we will discuss below). However, this theory-to-reality-gap is true for any provably correct/robust algorithm and not just ours. Still, our theoretical guarantee gives us more assurance of real-world system behavior compared to algorithms only evaluated empirically without any guarantees.
>
> **[What's the complexity of finding sensor events]** The reviewer is concerned that by assuming effective sensors are designed a priori by an oracle, we offload significant problem complexity to the oracle for hard imitation tasks. While this is true for tasks where even human demonstrators struggle to articulate the task success criteria, many practical problems considered by the LfD community [1] do have clear success criteria and thus the corresponding sensors can be designed with ease. For example, in our scooping task, we use a wrist-mounted camera to track if there is at least one red bead in the spoon at all times via color detection. For arbitrary objects, we could use the wrist-mounted depth sensor to check if there is anything on the spoon by depth thresholding. To detect if the spoon has reached an attractor, we track the spoon's distance to the bowl location. Note assuming these sensors can be designed and are given is a necessary concession, otherwise, the problem of certifying LfD methods to be invariant to modes of unknown boundaries will be intractable. Such an assumption is still significantly weaker and more realistic compared to the assumption made in TLP that the entire boundary of every mode must be known.
>
> **[How stable is the boundary]** The learned mode boundaries always form a convex inner-approximation of the true boundaries by construction (refer to QCQP optimization in appendix B.3), and are stable in the sense that they will not get refined unless new invariance failures are encountered. This guarantees that the modulation will be full rank and thus preserve the stability and convergence properties as proven in Theorem 1. The problematic oscillation across boundaries in the multi-modal experiments is due to the fact that vanilla DS is not mode invariant without modulation. LTL-DS with modulation is proved to not suffer from any such problematic oscillation while simply sequencing DS with LTL without modulation will.

---

> > ### Author Response · Authors · 2022-08-22
> > **Response to reviewer QTy6 (part 2)**
> >
> > **[Additional Tasks]** In the paper we have considered three categories of tasks: (1) Single-mode reaching tasks (section 6.1 and appendix B), (2) Multi-modal scooping tasks (section 6.2 and appendix C) (3) Multi-modal generalization tasks (section 6.2 and appendix D). The reviewer is concerned these tasks are not enough to illustrate the strength or limitations of LTL-DS. Admittedly, our contribution is not an extensive empirical evaluation of any particular LfD algorithm on a suite of tasks such as [1]. As our problem formulation has not been studied before and the theory is missing, we had only intended the empirical experiments in the paper to motivate and illustrate our theoretical contribution to the LfD community, i.e. a formal proof to certify LfD methods. That being said, the weakness of our theoretical result are the three assumptions made in the proof section 5 as well as the limitation section 7 (e.g., the assumption of having good sensors). Note if any of these assumptions is violated (e.g., if lighting affects the color detection accuracy of cameras), the theoretical success guarantee won't translate into a hardware experiment success guarantee. However, in this case, the failure is not due to our algorithm. We would make these limitations clearer in the revised draft. Furthermore, we are currently working on some additional robot demos per the reviewer's request and will include them in the appendix section in the next couple of days. Thanks for your patience!
> >
> > **[Additional Evaluations]** We very much agree with the reviewer that perturbations should come from unbiased sources. **We'd like to highlight that perturbations in the single mode reaching tasks are sampled from Gaussian noise (refer to Fig. 5 and Fig 8 in the original text and appendix B.1), which is indeed an unbiased source.** We are also recruiting more human subjects to perturb our physical setup and will publish results in the next couple of days. Thanks for your patience!
> >
> > [1] James, et al. RLBench: The Robot Learning Benchmark & Learning Environment

---

> > > ### Author Response · Authors · 2022-08-25
> > > **Response to reviewer QTy6: additional evaluations on the real robot**
> > >
> > > **Comment:**
> > >
> > > **[New results: additional human experiments]** Since external perturbations are an integral part of our task complexity, we recruited five human subjects without prior knowledge of our LTL-DS system to perturb the robot scooping setup. Each subject is given five trials of perturbations. In total, we collected 25 trials, each of which is seen as an unbiased i.i.d. source of perturbations. In our newly submitted video (which we also upload to the project page https://sites.google.com/view/ltl-ds), **we show all 25 trials succeed eventually. We did not cherry-pick the results and we shoot all videos in one go.** This empirical 100% success rate further corroborates our theoretic success guarantee. Interestingly, common perturbation patterns (we annotate with the same colored text) emerge from different participants. Specifically, we see adversarial perturbations where humans fight against the robot and collaborative perturbations where humans help the robot to achieve the goal of transferring at least one bead from one bowl to the other. In the case of adversarial perturbations, DS reacts and LTL replans. In the case of collaborative perturbations, DS is compliant and allows humans to also guide the motion. In the case where humans are not perturbing yet the robot makes a mistake (e.g. during scooping), LTL replans the scooping DS until the robot enters the transferring mode successfully. The fact that we don't need to hard code different rules to handle invariance failures caused by perturbations and the robot's own execution failures in the absence of perturbations highlights the strength of our LTL-powered sensor-based task reactivity. We will continue to present more robot experiments in the coming days.
> > >
> > > **Zip File:**
> > >
> > > /attachment/d70bf64515dc04d98a6f89dce2bcfc2d8f31e9ad.zip

---

> > > > ### Author Response · Authors · 2022-08-28
> > > > **Final response to reviewer QTy6**
> > > >
> > > > **Comment:**
> > > >
> > > > Dear reviewer,
> > > >
> > > > We are attaching three additional robot experiment videos to address your concern about too few tasks. Experiment 1 is explained in our previous response to all reviewers.  Experiment 2 is documented in the new Appendix J. Experiment 3 is testing different automaton structure, and we will include more details in the camera-ready version should the paper gets accepted. Thank you for your time and consideration!
> > > >
> > > >
> > > > **Zip File:**
> > > >
> > > > /attachment/909d347773fbc6d173a60f59a4b83e4fa42928da.zip

---

### Official Review · Reviewer_VK5r · 2022-08-06

**Originality:** Very Good
**Technical Quality:** Very Good
**Clarity Of Presentation:** Very Good
**Impact:** 4

**Recommendation:**

Weak Accept: I recommend accepting the paper, but will not argue for my recommendation if the majority of other reviewers have a different opinion.

**Summary:**

This work provides a temporal logic imitation method for robot operations, which can deal with both motion-level perturbations and task-level perturbations. In detail, a dynamic system is implemented along with collision boundaries for avoiding motion failure, and linear temporal logic is used for task-level failure detection and recovery. The experiments on real robots show the effectiveness of the proposed method, and the theoretical analysis of the dynamic system provides the guarantee of the G.A.S. of the DS and the satisfaction of the constraints (cuts).

**Issues:**

1. How to design and choose the binary sensor state in practice?

**Quality Of The Limitations Section:**

Limitations are addressed clearly

**Reviewer Expertise:**

3: The reviewer is fairly confident that the evaluation is correct

**Robotics Focus:**

Sufficient demonstration on hardware

**Strengths And Weaknesses:**

This work provides a robot skill imitation method with the ability to recover from failures and perturbations. The proposed method successfully combines DS and LTL and provides theoretical analysis on the guarantee of system reliability. The detained diagram that illustrates the principle of the proposed method is clear and intuitive, which helps the reader quickly understand the main idea of this work, and the video presentation is also very impressive.

However, as maintained by the author, this method itself relies on well-designed sensor states for successfully classifying the mode, which would cause some problems when applying to more complex robot tasks. Also, the experiments are mainly on the soup-scooping task, further evaluation on more complex tasks is welcomed.

**Summary Of Recommendation:**

This work is well organized and easy to read, the provided video presentation and diagram can illustrate the proposed method clearly and intuitively. The experiments on the soup-scooping task are a little bit simple but impressive, where the analysis of the results is sufficient and can well support the main claim of this work.

---

> ### Author Response · Authors · 2022-08-22
> **Response to reviewer VK5r**
>
> Dear reviewr VK5r,
>
> Thanks for your detailed review and positive evaluation! Please find our response below:
>
> **[How to design sensors in practice]** This work assumes sensors to detect mode transitions are available a priori. In many papers [1][2] (more in the related works section), the demonstrator usually knows what constraints are necessary for task success. For example, in our scooping task, we implemented a red bead detector on a wrist-mounted camera via color thresholding to track if there is at least one red bead in the spoon at all times. To detect if the spoon has reached an attractor, we track the spoon's distance to the bowl location. More details on sensor designs could be found in appendix E. We agree with the reviewer's assessment that imperfect sensors can complicate the learning of mode boundaries, but in practice, the aforementioned simple sensor designs prove very effective. In the future, we would like to investigate how sensor uncertainties can affect learning. Another interesting future direction mentioned in our limitation section would be to additionally learn the sensor state from demonstrations. However, this is out of scope as we want to focus on the introduction of the TLI problem and an initial solution to solve it. In the future, we plan to relax the assumption of oracle sensors made in the paper.
>
>
> **[Mainly one task]** We would like to clarify that while the soup-scooping task is the motivating example, we have three categories of different tasks testing different hypotheses in the paper. **Category 1: Single-mode reaching task. (original text section 6.1 and appendix B)** We investigate how the lack of bisimulation properties undermines the success guarantee for LfD methods. We show one task instance in the main paper and many more instances of the same task category in the appendix. **Category 2: Multi-modal scooping task. (section 6.2 and appendix C)** This task shows having a discrete reactive plan is not sufficient for success guarantee as the discrete plan might not be realizable by continuous LfD methods without the bisimulation property. This task is demonstrated both in simulation and on a real robot. **Category 3: Multi-modal generalization task. (section 6.2 and appendix D)** We show that policies learned from one task can generalize to other tasks with different temporal structures by updating the LTL formulas (no updates to the policies). We demonstrate the idea on variants of the "getting chicken and broccoli" task. Admittedly, our contribution is not an extensive empirical evaluation of any particular LfD algorithm on a suite of tasks such as [3]. As our problem formulation has not been studied before and the theory is missing, we had only intended the empirical experiments in the paper to motivate and illustrate our theoretical contribution to the LfD community, i.e. a formal proof to certify LfD methods. Lastly, we are currently working on a second robot experiment involving a completely different task, sensor states, and LTL formula, which we will publish soon. Thanks for your patience!
>
> **[Task is too simple]** The scooping task in our paper in fact consists of three subtasks: scooping, transporting (i.e., constraint following), and pouring, each of which has been studied as a standalone task in some papers [4]. Nevertheless, we agree that our task might still be simple compared to tasks in [1][5][6] if **task complexity is measured in time-horizon**. But these long-horizon tasks are all prehensile pick-and-place tasks, so objects once grasped are assumed to stay in hand (no unexpected mode transitions). In contrast, our scooping task is non-prehensile, and scooped objects can be dropped and lead to unexpected mode transitions, resulting in more complexity at the planning level. Thus, if **task complexity is measured in terms of task structure**, our scooping task is more complex than prehensile tasks with a linear task structure (i.e., automaton as a linear chain of nodes) even though they might have a longer time horizon. Furthermore, arbitrary perturbations are also an integral part of the task design. Tasks in [1][3][4][6] are evaluated without external perturbations, let alone a success guarantee. We believe **external perturbation is also an important dimension of the task complexity**, and we hope this work could help start a debate on what makes a task complex in the LfD community.
>
> [1] Niekum, et al. Incremental semantically grounded learning from demonstration.
>
> [2] Perez-D’Arpino, Shah. C-LEARN: Learning Geometric Constraints from Demonstrations for Multi-Step Manipulation in Shared Autonomy
>
> [3] James, et al. RLBench: The Robot Learning Benchmark & Learning Environment
>
> [4] Bahl, et al. Hierarchical Neural Dynamic Policies
>
> [5] Lee, et al. IKEA Furniture Assembly Environment for Long-Horizon Complex Manipulation Tasks
>
> [6] Gupta, et al. Relay Policy Learning: Solving Long-Horizon Tasks via Imitation and Reinforcement Learning

---

> > ### Author Response · Authors · 2022-08-28
> > **Final response to reviewer VK5r**
> >
> > **Comment:**
> >
> > Dear reviewer,
> >
> > We are attaching three additional robot experiment videos to address your concern about too few tasks. Experiment 1 is explained in our previous response to all reviewers.  Experiment 2 is documented in the new Appendix J. Experiment 3 is testing different automaton structure, and we will include more details in the camera-ready version should the paper gets accepted. Thank you for your time and consideration!
> >
> >
> > **Zip File:**
> >
> > /attachment/1a283d5ba28c5f3753bdd5a9f4f4ac1ddde2cd60.zip

---

### Meta-Review · Area_Chair_qD2D · 2022-08-07

**Recommendation:** Accept (Oral)
**Confidence:** 4

**Metareview:**


I invite the authors to respond to all issues and questions by the reviewers and make modifications accordingly.

Excerpt of the reviews:

Strengths:
* (OTy6) The community is in need of methods that move away from “just” action imitation into some forms of imitation with better abstractions and objectives
* (QTy6) The combination of control with convergence warranties with hybrid automata is novel and interesting for the community.
* (all) real robot experiments

Weaknesses / Issues:
* (QTy6) Make clear what is prior work and what is a novel contribution, possibly with a "Preliminaries" section
* (QTy6) Additional tasks and evaluation conditions on the real robot
* (VK5r, L7Wt) mainly one task: soup-scooping -> more tasks would be better
* (p7Tc) little to no relation to learning
* (p7Tc) not enough details (supply more information in the appendix)
----- post rebuttal
The authors have addressed weaknesses by improving the relation to prior work, adding more (unbiased) evaluation, and adding 2 different tasks.
The paper should definitely be accepted and can be considered for an oral presentation (weak endorsement).

I ask the authors to incorporate the changes into the main text as much as possible and make references to the new material in the supplementary at the appropriate places.

---

> ### Author Response · Authors · 2022-08-22
> **General response to all reviewers**
>
> We appreciate the meta-reviewer's summary of strengths and weaknesses, and we thank all reviewers for their invaluable feedback! We found the following common questions and would like to address them to all:
>
> **[Contribution of this work]** We uploaded a supplementary file including (1) a re-write of sections 3-5 into preliminaries and additional details on our approach for clarification (If accepted we will incorporate these changes to the main text), and (2) a new appendix H "Relation of TLI to Prior Work", where we explain how our work and prior work fit together.
>
> **[Mainly one task]** As explained in the detailed response to reviewer VK5r, we have evaluated three task categories in simulation in addition to the real robot experiment. Details can be found in Appendix B, Appendix C, Appendix D, and Appendix E. Additionally, we are working on a second robotic experiment that involves a completely different task, sensor states, LTL formula, etc, which we will be uploading soon. Admittedly, our contribution is not an extensive empirical evaluation of an LfD algorithm on a suite of tasks such as [2]. As our problem formulation has not been studied before and the theory is missing, we had only intended the empirical experiments in our paper to motivate and illustrate our **main theoretical contribution to the LfD community, i.e. a novel problem formulation and a formal proof to certify LfD methods.**
>
>
> **[Task is too simple]** Many reviewers found the scooping task too simple although impressive. As explained in the detailed response to reviewer VK5r, the scooping task is simple only if task complexity is measured in time-horizon alone. If the capability to replan on the fly according to a non-linear automaton structure and the capacity to handle arbitrary perturbations are considered part of the task complexity, our scooping task is more complex than the many tasks presented in [1][2][3][4]. The fact we don't know any algorithm prior to this work that can provably succeed in our scooping task (or in general tasks with temporal modal constraints) given **arbitrary external perturbations** is an indication that the task is, contrary to our intuition, not that simple after all. We hope this work could lead the LfD community to consider what other aspects of a task (other than its time horizon or the number of subgoals in a linear sequence) make the task complex.
>
> **[little to no relation to learning]** The primary motivation for reviewer p7Tc to reject this paper despite the reviewer's positive evaluation in many metrics is the little relevance to learning. In our response to reviewer p7Tc, we argue our paper has satisfied the CoRL in-scope criteria of demonstrating relevance to learning either through intent or outcome. In fact, reviewer L7Wt made the comment in "Summary of Recommendation" that this work "is well placed within the existing LfD literature." We hope our newly submitted appendix H "Relation of TLI to Prior Work" as well as the re-write of sections 3-5 will clarify the learning aspects of this paper. Specifically, LTL-DS is a form of imitation learning augmentation defined as “Teacher-Student Interaction during Behavioral Cloning” in [5].
>
>
> [1] Niekum, et al. Incremental semantically grounded learning from demonstration.
>
> [2] James, et al. RLBench: The Robot Learning Benchmark & Learning Environment
>
> [3] Lee, et al. IKEA Furniture Assembly Environment for Long-Horizon Complex Manipulation Tasks
>
> [4] Gupta, et al. Relay Policy Learning: Solving Long-Horizon Tasks via Imitation and Reinforcement Learning
>
> [5] Osa, et al. An Algorithmic Perspective on Imitation Learning

---

> > ### Author Response · Authors · 2022-08-25
> > **General response to all reviewers: additional evaluations on the real robot**
> >
> > **Comment:**
> >
> > **[New results: additional human experiments]** Since external perturbations are an integral part of our task complexity, we recruited five human subjects without prior knowledge of our LTL-DS system to perturb the robot scooping setup. Each subject is given five trials of perturbations. In total, we collected 25 trials, each of which is seen as an unbiased i.i.d. source of perturbations. In our newly submitted video (which we also upload to the project page https://sites.google.com/view/ltl-ds), **we show all 25 trials succeed eventually. We did not cherry-pick the results and we shoot all videos in one go.** This empirical 100% success rate further corroborates our theoretic success guarantee. Interestingly, common perturbation patterns (we annotate with the same colored text) emerge from different participants. Specifically, we see adversarial perturbations where humans fight against the robot and collaborative perturbations where humans help the robot to achieve the goal of transferring at least one bead from one bowl to the other. In the case of adversarial perturbations, DS reacts and LTL replans. In the case of collaborative perturbations, DS is compliant and allows humans to also guide the motion. In the case where humans are not perturbing yet the robot makes a mistake (e.g. during scooping), LTL replans the scooping DS until the robot enters the transferring mode successfully. The fact that we don't need to hard code different rules to handle invariance failures caused by perturbations and the robot's own execution failures in the absence of perturbations highlights the strength of our LTL-powered sensor-based task reactivity. We will continue to present more robot experiments in the coming days.
> >
> > **Zip File:**
> >
> > /attachment/1d681f6bf10500e14deaca629a5834b6c5f08dc0.zip

---

> > > ### Author Response · Authors · 2022-08-28
> > > **Final response to meta reviewer**
> > >
> > > **Comment:**
> > >
> > > Dear reviewer,
> > >
> > > We have addressed all other reviewers' questions. We are also attaching three additional robot experiment videos to address the common concern about too few tasks. Experiment 1 is explained in our previous response to all reviewers. Experiment 2 is documented in the new Appendix J. Experiment 3 is testing different automaton structure, and we will include more details in the camera-ready version should the paper gets accepted. Thank you for your time and consideration!
> > >
> > >
> > >
> > > **Zip File:**
> > >
> > > /attachment/cb32cc5caa53d57e9341380b40c0fddb0ffde087.zip